# Prenatal Exposure to Metals and Neurodevelopment in Infants at Six Months: Rio Birth Cohort Study of Environmental Exposure and Childhood Development (PIPA Project)

**DOI:** 10.3390/ijerph19074295

**Published:** 2022-04-03

**Authors:** Mônica Seefelder de Assis Araujo, Carmen Ildes Rodrigues Froes-Asmus, Nataly Damasceno de Figueiredo, Volney Magalhães Camara, Ronir Raggio Luiz, Arnaldo Prata-Barbosa, Marlos Melo Martins, Silvana do Couto Jacob, Lisia Maria Gobbo dos Santos, Santos Alves Vicentini Neto, Jorge Fonte de Rezende Filho, Joffre Amim Junior

**Affiliations:** 1Public Health Institute, Federal University of Rio de Janeiro, Rio de Janeiro 21941-592, Brazil; volney@iesc.ufrj.br (V.M.C.); ronir@iesc.ufrj.br (R.R.L.); 2Postgraduate Program in Perinatal Health, Faculty of Medicine, Maternity School of Rio de Janeiro, Federal University of Rio de Janeiro, Rio de Janeiro 22240-000, Brazil; carmenfroes@iesc.ufrj.br (C.I.R.F.-A.); natalydamasceno@iesc.ufrj.br (N.D.d.F.); arnaldoprata@outlook.com (A.P.-B.); rezendef@me.ufrj.br (J.F.d.R.F.); joffre@me.ufrj.br (J.A.J.); 3D’Or Institute for Research & Education (IDOR), Rio de Janeiro 22281-100, Brazil; 4School Maternity Hospital, Federal University of Rio de Janeiro, Rio de Janeiro 22240-000, Brazil; marlosmartins@me.ufrj.br; 5Oswaldo Cruz Foundation, National Institute of Quality Control in Health, Rio de Janeiro 21040-900, Brazil; silvana.jacob@incqs.fiocruz.br (S.d.C.J.); lisia.gobbo@incqs.fiocruz.br (L.M.G.d.S.); santos.neto@incqs.fiocruz.br (S.A.V.N.)

**Keywords:** prenatal exposures, children’s health, heavy metals, neurodevelopment, neurodevelopmental screening, umbilical cord blood

## Abstract

The PIPA Project is a prospective birth cohort study based in Rio de Janeiro, Brazil, whose pilot study was carried out between October 2017 and August 2018. Arsenic (As), cadmium (Cd), lead (Pb), and mercury (Hg) concentrations were determined in maternal (*n* = 49) and umbilical cord blood (*n* = 46). The Denver Developmental Screening Test II (DDST-II) was applied in 50 six-month-old infants. Metals were detected in 100% of the mother and newborn samples above the limits of detection. Maternal blood lead concentrations were higher in premature newborns (GM: 5.72 µg/dL; *p* = 0.05). One-third of the infants (*n* = 17–35.4%) exhibited at least one fail in the neurodevelopment evaluation (fail group). Maternal blood arsenic concentrations were significantly (*p* = 0.03) higher in the “fail group” (GM: 11.85 µg/L) compared to infants who did not fail (not fail group) (GM: 8.47 µg/L). Maternal and umbilical cord blood arsenic concentrations were higher in all Denver Test’s domains in the “fail group”, albeit non-statistically significant, showing a tendency for the gross motor domain and maternal blood (*p* = 0.07). These findings indicate the need to further investigate the toxic effects of prenatal exposure to metals on infant neurodevelopment.

## 1. Introduction

Metals are significant environmental pollutants and may originate from natural sources or as a result of human activities [1]. The Agency for Toxic Substances and Disease Registry (ATSDR) lists arsenic (As), cadmium (Cd), lead (Pb), and mercury (Hg) in the Priority Substance List, based on a combination of frequency, toxicity, and potential for human exposure [2].

Chronic low-dose exposure to these metals may cause toxic human health effects affecting the nervous, hematological, and immune systems [3]. Environmental exposure to heavy metals can occur through the inhalation (indoor and outdoor air pollution) or ingestion of contaminated food and water [4,5]. Some metals, such as mercury and lead, are known neurotoxic elements [6]. Neurotoxic metals may represent a risk for child neurodevelopment during critical early life periods [7]. For example, fetal brain development during pregnancy is susceptible to the action of neurotoxic substances [6], as the placental barrier is not completely impermeable to the passage of harmful substances, including heavy metals [8]. Several studies have reported an association between metal exposure during pregnancy and early childhood and impaired cognitive function in children [9,10,11,12].

Neurobehavioral developmental disorders such as autism spectrum disorder, attention deficit hyperactivity disorder, and subclinical decreases in brain function appear to be on the increase worldwide [13]. According to Dórea (2020) [7], in Latin American and Caribbean (LAC) countries, there are more than 300 million children under threat of neurodevelopmental delays due to environmental exposure to neurotoxic substances, including metals. The etiology of metals in neurodevelopmental disorders can occur through different pathways that include genetic and epigenetic mechanisms, neuroendocrine dysfunction, oxidative stress, immune dysregulation, and changes in neurotransmitters [14].

In Brazil, few studies have investigated metals in maternal and umbilical cord blood [15,16,17,18,19]. In one study carried out with 117 newborns in the city of Rio de Janeiro, Southeastern Brazil, de Assis Araujo et al. (2020) [19] observed As, Cd, Pb, and Hg concentrations above the limit of detection (LD) in all maternal and umbilical cord blood samples. The authors referred that 50% of the investigated newborns presented Pb concentrations above 3.5 μg/dL, which has been the CDC blood level reference value since May 2021 [20], and for mercury, no newborn presented concentrations above of 5.8 μg/L, which is the cord blood limit established by some authors [21,22]. Some studies have assessed exposure to mercury, lead, and aluminum in the neurodevelopment of children younger than 5 in the Amazon region and have suggested an association between high mercury exposure and poorer performance on neurobehavioral tests [23]. As far as we are aware, there are no studies in Rio de Janeiro investigating the concentrations of metals in maternal blood and in the umbilical cord and neuromotor development.

The Rio Birth Cohort Study on Environmental Exposure and Child Development (PIPA Project) is a prospective cohort study conducted at the Federal University of Rio de Janeiro Maternity School, in the city of Rio de Janeiro, southeastern Brazil, aiming at the investigation of the effects of environmental pollutants on maternal and child health. A pilot study (PIPA Pilot study) was carried out between September 2017 and August 2018, with 142 pregnant women enrolled in and 135 children born at the Federal University of Rio de Janeiro Maternity School [24]. This study investigates potential associations between metal concentrations in maternal and umbilical cord blood and newborn neurological development among the PIPA pilot study population.

## 2. Materials and Methods

### 2.1. Study Area

The PIPA Pilot Study was carried out for 12 months, from September 2017 to August 2018, at the UFRJ Maternity School (UFRJ Maternity). The UFRJ Maternity is located in the south zone of the city of Rio de Janeiro. It is one of the public hospitals of Rio de Janeiro responsible for prenatal assistance and delivery to all pregnant women living in the city, including those with high-risk pregnancies.

### 2.2. Study Population

A total of 135 children were born at the UFRJ Maternity between October 2017 and February 2018 during the PIPA Pilot Study. Five dropouts occurred, so 130 newborns were eligible for follow-up visits. The study population comprised the children that attended the follow-up visit in the sixth month. Exclusion criteria were children admitted to the intensive care unit during the neonatal period, birth weight <2.5 kg, gestational age <34 weeks, and mother drugs use during pregnancy.

The infants were evaluated using the Denver Developmental Screening Test II (DDST-II). Children whose parents said they had not had the opportunity to experience the compliance of the DDST-II task were scored as “no opportunity” and were excluded.

The study population was 48 children (Figure 1).

### 2.3. Arsenic, Cadmium, Lead, and Mercury Determinations

Arsenic, cadmium, lead, and mercury in maternal blood samples collected during the third trimester of pregnancy and umbilical cord blood collected during delivery were analyzed employing the inductively coupled plasma mass spectrometry technique (ICP-MS), at the National Institute for Quality Control in Health (INCQS) laboratory, at the Oswaldo Cruz Foundation (FIOCRUZ). The limits of quantification (LQ) for As, Cd, Pb, and Hg were, respectively 0.01 μg/L, 0.006 μg/L, 0.05 μg/L, and 0.02 μg/L, while the limits of detection (LD) were 0.003 μg/L, 0.002 μg/L, 0.015 μg/L, and 0.007 μg/L, respectively. The sample collection and analysis procedures have been previously described by Asmus et al. (2020) [24].

### 2.4. Denver Development Screening Test II (DDST-II)

Infant neurodevelopment was assessed using the Denver Developmental Screening Test II (DDST-II). The DDST-II is the 1992 revised version of the Denver Screening Test developed by Frankenburg and Dodds in 1967 [25]. The DDST-II used herein is the version translated into Portuguese. The Brazilian Society of Pediatrics recommends this test to monitor child development [26]. The DDST-II can be applied by health professionals directly to the child or through its parents or guardians. The purpose of this tool is to screen neuromotor development, evaluating children from zero to six years old regarding their ability to perform tasks organized in four neurodevelopmental domains: “Personal-Social”, “Fine Adaptive Motor”, “Gross Motor”, and “Language”. Each task is represented by a bar that indicates the age in which task compliance is performed by 25%, 50%, 75%, and 90% of children. A score is given to each task evaluated, as follows: “pass, fail, refuse, no opportunity” [27].

For all newborns, a follow-up appointment at sixth months was scheduled, when the DDST-II was applied for neurodevelopment assessment, in the presence of the mother by a specially trained examiner who was unaware of the child’s developmental history and of the metal levels at the time of the test.

For the analyses, the children were divided into “fail group” and “not fail group”. Following the guidance of the DDST II manual, “fail” and “refuse” scores were considered as “fail”, and “pass” scores were considered as “not fail” [27]. 

### 2.5. Statistical Analyses

A descriptive analysis of participants characteristics was performed. Frequency distributions for categorical variables and arithmetic mean and median for the continuous variables were calculated. The geometric means, 25th, 50th, 75th, 90th, and 95th percentiles, and the minimum and maximum values of the pollutant concentration levels were calculated for the maternal and umbilical cord blood samples.

Spearman’s correlation or non-parametric comparisons (Mann–Whitney-U-test) or parametric (Student) tests were used to evaluate continuous variables, and Chi-squared or Fisher tests were used for categorical variables.

To identify confounders and make adjustments, we evaluated the association between metals and maternal and infant sociodemographic characteristics. We also verified the relationship between maternal sociodemographic characteristics and characteristics of birth with the DDST-II outcome. To select which covariates to adjust, we set a *p*-value < 0.1. No covariates were associated with exposure and outcome simultaneously, so we did not use a regression model to adjust for confounders. All data were analyzed using the Statistical Package for Social Science v.21 software for Windows.

### 2.6. Covariates

The continuous variables—mother age, gestational age, birth weight, per capita income, and education—were counted in years. 

Categorical variables were ethnicity, gender, alcohol consumption, tobacco exposure, Apgar index, adequacy for gestational age, and prematurity.

Maternal sociodemographic data included maternal age, per capita income, education and men, ethnicity, exposure to tobacco, and alcohol consumption, which were collected through questionnaires administered by trained interviewers.

Birth information was obtained from the birth record.

Family neurological history for autism and ADHD was revealed by the mother at the follow-up visit, but there was no report of these disorders in parents or siblings.

## 3. Results

Arsenic, Cd, Pb, and Hg were above the LQ in all maternal blood and umbilical cord blood samples (Table 1). A significant positive correlation was observed between all metal concentrations in maternal and umbilical cord blood, as previously described by Figueiredo et al. (2020) [28].

Mean mother age was 28.54 years (SD ± 7.07), with a mean per capita income of USD 257.00 (SD ± 167.38) and median of 14 years of schooling; 71.4% of the mothers were non-white, 36.7% reported exposure to tobacco (personal and second-hand exposure), and 50.0% consumed alcohol during pregnancy. A statistically significant positive correlation between maternal age and cord blood concentrations were observed for Hg (r = 0.34; 0.02) and Pb (r = 0.29; 0.05). Non-white pregnant women presented higher geometric Cd means in maternal blood (*p* = 0.04) and cord blood (*p* = 0.08) compared to white women (Table 2).

Mean birth weight was 3.3 kg (SD ± 0.5 kg) and median gestational age was 39 weeks (range: 36–41 weeks), 62.0% (*n* = 31) of all infants were male, 11.1% (*n* = 5) were small for gestational age (SGA), and 6.0% (*n* = 3) were premature (IG 34–37 weeks). The geometric Pb mean in maternal blood (5.72 μg/dL; *p* = 0.05) and in cord blood (7.23 μg/dL; *p* = 0.16) was higher in premature babies. An inverse correlation trend was observed between gestational age and Cd concentrations in maternal blood (−0.25; *p* = 0.08) and in umbilical cord blood (−0.21; *p* = 0.16) (Table 3).

In the sixth month follow-up visit, 48 eligible children were evaluated with regard to the four neurodevelopmental Denver Test’s domains. Most (31–64.6%) did not fail in any domain (“not fail” group), and 17 (35.4%) children failed in one or more domains (“fail” group). No differences were observed concerning maternal sociodemographic characteristics and birth characteristics between the “fail” and “not fail” groups (Table 4).

Comparisons between the metal concentrations geometric means in maternal blood and cord blood between the “fail” and “not fail” groups in the sixth month are displayed in Table 5. The geometric means for maternal blood arsenic were significantly higher in the “fail group” compared to the “not fail group” (*p* = 0.03). The maternal and cord blood arsenic concentrations were higher also in the “fail group” for the personal social (*p* = 0.74), fine adaptative motor (*p* = 0.27), language (*p* = 0.36), and gross motor (*p* = 0.07) Denver Test’s domains, (Table 6), albeit non-significantly. No significant results were observed for the other metals between the “fail” and “not fail” groups and each DDST-II domain.

## 4. Discussion

To the best of our knowledge, this study is the first in the city of Rio de Janeiro to investigate the relation between prenatal exposure to As, Cd, Pb, and Hg and the neurodevelopment of children up to 6 months of age.

In our study, 34.5% of children (*n* = 17) had at least one failure in the DDST-II. In another study carried out in Acre (Brazil) with 47 children up to 12 months of age, 72.7% of children between 4 and 6 months of age (*n* = 3) had less than two failures, which were results considered adequate for the age group [29]. The difference between our study and the one carried out by Andrade et al. (2013) [29] is probably due to the difference in the age range of the population at risk and in the number of failures considered adequate. In the study carried out by Marques et al. (2007) [30], 74% of the babies evaluated at 6 months of age by the Gesell Developmental Studies showed results considered adequate for their age group. Although the age of the study population was the same as ours, the test to assess neurodevelopment was different.

Birth conditions and nutritional status are known to interfere in child neurodevelopment [31,32,33,34]. Halpern et al. (1996) [31], in a study carried out in the city of Pelotas, southern Brazil, with 1362 one-year-old children, observed the occurrence of neurodevelopmental failures through the application of the DDST-II, which was inversely associated with birth weight. Higher birth weights were observed herein in the “not fail” group, although they were not statistically significant. No significant associations were observed between mother age, income, education, ethnicity, or tobacco or alcohol use and the “fail” and “not fail” groups. The differences between the results reported herein and other studies may be due to sample size, metal concentrations, the type of test used to assess neurodevelopment, and the influence of sociodemographic factors and lifestyle [35].

A significant positive correlation was observed between As, Cd, Pb, and Hg concentrations in maternal and umbilical cord blood, as previously described by Figueiredo et al. (2020) [28]. The higher Hg and Pb concentrations in umbilical cord compared to maternal blood can be attributed to increased glomerular filtration rates during pregnancy, resulting in higher renal elimination [36].

In this study, maternal and umbilical cord blood geometric means for As were higher in the “fail” group for DDST-II in the six months. This result was more significant for As maternal blood (*p* = 0.03) and the gross motor domain (*p* = 0.07). Some studies have investigated the effect of prenatal exposure to As on neurological infant development. Liang et al. (2020) [37], in a prospective Chinese birth cohort study (Ma’anshan Birth Cohort-MABC) on 2315 six-month-old infants, observed associations between umbilical cord serum As concentration (median = 1.89 μg/L) and suspected developmental delay in the personal–social domain. Wang et al. (2018) [38], in a cross-sectional study carried out in Shanghai (*n* = 892), observed that newborns with a low neonatal behavioral neurological assessment (NBNA) score exhibited higher concentrations (median, interquartile range) of umbilical cord As (3.93, 1.27–8.21 μg/L) than those with a high NBNA score (0.61, 0.23–1.50 μg/L). Tolins et al. (2014) [39] concluded in a review study that intrauterine exposure may be associated with neurodevelopmental deficits, even at exposure levels below current safety guidelines (10 µg/L in drinking water), with manifestations detected only later in life. Those authors also observed that certain factors, such as sex, concomitant exposures, and exposure time, modify developmental As neurotoxicity. In our study, we found that As was not associated with maternal sociodemographic conditions or the baby’s birth conditions.

In our study, there was no significant difference in maternal blood cadmium concentrations between the fail and not fail groups. Review studies reported that the effects of prenatal exposure to cadmium and childhood neurodevelopment are still not clear, and there is still no consensus about the toxic effect of this metal on cognitive performance [40,41].

The maternal and cord blood geometric means for Cd were higher in the non-white ethnic group, which was significant only for maternal blood. Some studies suggest that higher biological Cd levels may be associated to a more socioeconomically vulnerable population, including socioeconomically less favored ethnicities [42,43].

Numerous studies support the concept that Pb affects cognitive function in children exposed prenatally and/or environmentally to low levels of Pb and that no safe parameter has been identified [44]. No associations were observed between failures in the neurodevelopment domains and maternal or umbilical cord Pb concentrations in our study population.

In general, the literature points to a more expressive toxic effect of lead exposure on the child neurodevelopment than related to the other metals exposure. One issue to be observed is that the As geometric mean (11.48 µg/L) is much higher than the Pb geometric mean (4.44 µg/dL) in the Fail group. It could be a reason, but the small size of the group does not permit any conclusion.

The maternal blood Pb concentrations were significantly higher in preterm infants (gestational age >34 weeks and <37 weeks). One hypothesis of the mechanism of action of lead in prematurity is through oxidative stress [45,46].

Cord blood lead was positively correlated with maternal age. Other studies have also found that maternal age was associated with higher levels of cord blood lead [47,48]. This association may be related to the accumulation of this metal in the body with age and the placental transport of Pb from the mother to the fetus [49].

No associations were observed between failures in the neurodevelopment domains and maternal or umbilical cord Hg concentrations in our study population. Nevertheless, in a systematic review study, Asmus et al. (2016) [23] reported a body of studies demonstrating associations between high child exposure to Hg in the Brazilian Amazon and poor neurobehavioral assessment results. Marques et al. (2007, 2009) [49,50] evaluated the neuromotor development of 82 six-month-old children in the Brazilian Amazon and observed an inverse and significant correlation between hair Hg contents and neurodevelopment delay (r = −0.333; *p* = 0.002).

Mercury in umbilical cord blood was positively correlated with maternal age. This correlation can be explained by the bioaccumulation of this metal in the maternal organism and by its transfer across the placental barrier [8,51].

## 5. Strengths and Limitations of This Study

The strength of this study is to provide information on prenatal environmental exposure to metals and potentially adverse effects on neurodevelopment at six months of age, covering sociodemographic and lifestyle data.

The limitations of this study are the small number of participants and the one-time neurological assessment. Many factors can interfere in the child’s process of neurodevelopment, and a long-term evaluation, with monitoring of the multiple intervenient factors, is the best proceeding.

## 6. Conclusions

This study shows that higher concentrations of arsenic in maternal blood were found in the group of children who failed in DDST-II. Continuous monitoring of concentrations of neurotoxic substances in women of childbearing age is essential to establish preventive measures to eliminate or minimize the risk of fetal exposure during pregnancy. Long-term development studies may reveal other associations. Future investigations should aim to demonstrate the possible neurodevelopmental effects of prenatal exposure to environmental pollutants on child health and identify potential sources of exposure.

## Figures and Tables

**Figure 1 ijerph-19-04295-f001:**
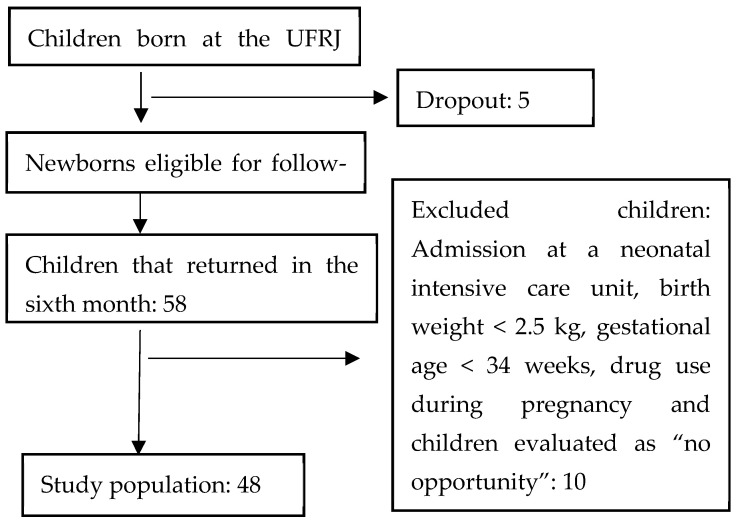
Selection process of the study population.

**Table 1 ijerph-19-04295-t001:** Descriptive statistics for arsenic, cadmium, lead, and mercury concentrations in maternal and umbilical cord blood, Rio de Janeiro, Brazil, 2018.

Metal	Sample	GM (95% CI)	Min.	P25	Median	P75	P90	P95	Max	R * (*p*-Value)
Arsenic (µg/L)	mb	9.46 (7.61–11.18)	0.33	8.27	9.89	11.79	17.47	19.21	36.48	0.87 (<0.001)
	ucb	10.07 (9.17–10.98)	5.06	8.29	10.27	12.06	15.37	16.95	19.94	
Cadmium (µg/L)	mb	0.29 (0.18–0.43)	0.01	0.13	0.30	1.29	2.09	4.74	9.97	0.76 (<0.001)
	ucb	0.32 (0.21–0.46)	0.01	0.15	0.33	0.83	2.33	4.40	4.88	
Lead (µg/dL)	mb	3.83 (3.32–4.43)	1.32	2.54	4.41	5.55	7.11	10.26	12.41	0.78 (<0.001)
	ucb	3.81 (3.81–4.53)	1.43	2.42	3.39	4.90	13.42	14.62	16.03	
Mercury (µg/L)	mb	0.90 (0.74–1.11)	0.38	0.61	0.76	1.42	2.77	3.26	13.32	0.64 (<0.001)
	ucb	0.95 (0.82–1.11)	0.42	0.69	0.88	1.30	2.64	3.42	4.52	

Maternal blood *n* = 49 and cord blood *n* = 46; GM = geometric mean; CI = confidence interval; mb = maternal blood; ucb = umbilical cord blood; * Spearman’s correlation; metal limits of quantification: arsenic 0.01 μg/L; cadmium 0.006 μg/L; lead 0.005 μg/dL; mercury Hg 0.02 μg/L.

**Table 2 ijerph-19-04295-t002:** Arsenic, cadmium, lead, and mercury concentrations in maternal blood and umbilical cord blood and maternal sociodemographic characteristics, Rio de Janeiro, Brazil, 2018.

	Arsenic µg/L				Cadmium µg/L				Lead µg/dL				Mercury µg/L			
	Maternal Blood		Cord Blood		Maternal Blood		Cord Blood		Maternal Blood		Cord Blood		Maternal Blood		Cord Blood	
R (*p*-value) ^a^																
Mother age (years)																
28.54 ± 7.07 *	−0.10 (0.51)		0.09 (0.54)		−0.05 (0.71)		0.12 (0.45)		0.15 (0.29)		0.29 (0.05)		0.17 (0.25)		0.34 (0.02)	
Per capita income (US$)																
257 ± 167.38 *	−0.10 (0.54)		0.06 (0.70)		−0.12 (0.45)		0.04 (0.78)		−0.21 (0.18)		−0.07 (0.67)		−0.07 (0.64)		0.18 (0.26)	
Education (years)																
14 (5–19) **	−0.01 (0.93)		0.01 (0.93)		−0.05 (0.75)		0.17 (0.27)		−0.14 (0.35)		0.07 (0.65)		−0.17 (0.23)		−0.06 (0.68)	
GM *p*-value ^b^ (CI 95%)																
Ethnicity																
Non-white— 71.4% (35) ***	9.41 (7.07–11.15)	0.84	10.39 (9.22–11.48)	0.55	0.37 (0.21–0.66)	0.04	0.39 (0.23–0.63)	0.08	3.73 (3.15–4.41)	0.67	3.87 (3.19–4.81)	0.96	0.94 (0.74–1.24)	0.78	0.98 (0.83–1.19)	0.40
White— 28.6% (14) ***	9.96 (7.89–12.54)		9.80 (8.13–11.92)		0.15 (0.09–0.25)		0.19 (0.10–0.37)		3.97 (2.98–5.05)		3.69 (2.63–5.17)		0.82 (0.61–1.13)		0.91 (0.66–1.28)	
Tobacco exposure																
No—63.3% (31) ***	10.62 (8.92–12.72)	0.58	10.66 (9.30–12.15)	0.12	0.30 (0.16–0.56)	0.72	0.37 (0.24–0.57)	0.69	4.09 (3.34–9.99)	0.24	3.96 (3.21–4.79)	0.30	0.96 (0.75–1.27)	0.44	0.90 (0.75–1.09)	0.78
Yes—36.7% (18) ***	9.59 (8.33–11.14)		9.31 (8.28–10.47)		0.27 (0.14–0.50)		0.37 (0.24–0.57)		3.50 (2.89–4.24)		3.61 (2.65–5.15)		0.84 (0.64–1.12)		1.03 (0.80–1.36)	
Alcohol consumption																
No—50.0% (25) ***	8.65 (5.78–11.95)	0.16	9.98 (8.68–11.36)	0.50	0.24 (0.11–0.51)	0.79	0.32 (0.17–0.59)	0.88	3.67 (2.93–4.63)	0.66	3.62 (2.90–4.57)	0.39	0.84 (0.62–1.26)	0.40	0.89 (0.76–1.06)	1.00
Yes—50.0% (25) ***	10.24 (8.78–11.90)		10.15 (8.90–11.48)		0.34 (0.20–0.57)		0.32 (0.19–0.54)		3.97 (3.37–4.69)		3.98 (3.06–5.17)		0.95 (0.77–1.20)		1.01 (0.78–1.32)	

Maternal blood *n* = 49 and cord blood *n* = 46; * arithmetic mean (±SD); ** median minimum–maximum; *** frequency (count); GM = geometric mean; CI = confidence interval = Spearman’s correlation; ^a^ Spearman’s *p*-value; ^b^ U Mann–Whitney test *p*-value.

**Table 3 ijerph-19-04295-t003:** Descriptive characteristics of birth outcomes and correlations with metal concentrations in maternal blood and umbilical cord blood, Rio de Janeiro, Brazil, 2018.

	Arsenic µg/L				Cadmium µg/L				Lead µg/dL				Mercury µg/L			
	Maternal Blood		Cord Blood		Maternal Blood		CORD BLOOD		Maternal Blood		Cord Blood		Maternal Blood		Cord Blood	
R (*p*-value) ^a^																
Birth weight (kg)																
3.3 ± 0.5 *	0.08 (0.61)		0.09 (0.54)		0.01 (0.96)		0.06 (0.71)		0.00 (1.00)		0.04 (0.79)		0.16 (0.28)		0.05 (0.75)	
Gestational age (weeks)																
39 (36–41) **	0.12 (0.43)		0.03 (0.84)		−0.25 (0.08)		−0.21 (0.16)		−0.19 (0.19)		−0.18 (0.23)		−0.19 (0.19)		−0.24 (0.10)	
GM *p*-value ^b,c^ (95% CI)																
Gender																
Male 62.0% (31) ***	10.39 (9.10–11.75)	0.33	10.26 (9.21–11.35)	0.39	0.27 (0.16–0.46)	0.75	0.32 (0.19–0.54)	0.91	3.51 (2.93–4.24)	0.16	3.59 (2.86–4.65)	0.36	0.94 (0.73- 1.23)	0.49	0.96 (0.77–1.18)	0.82
Female 38.0% (19) ***	8.11 (4.85–11.98)		9.77 (8.12–11.69)		0.32 (0.14–0.72)		0.31 (0.17–0.62)		4.41 (3.56–5.46)		4.20 (3.30–5.48)		0.83 (0.63–1.14)		0.94 (0.76–1.16)	
Apgar at 5 min																
≥8 98.0% (49) ***	9.46 (7.79- 11.19)	0.14	10.07 (9.17–11.02)	-	0.29 (0.18–0.44)	0.29	0.32 (0.21–0.47)	-	3.83 (3.33–4.40)	0.67	3.81 (3.21–4.55)	-	0.90 (0.75–1.12)	0.26	0.95 (0.82–1.12)	-
<8 2.0% (1) ***^d^	-		-		-		-		-		-		-			
Preterm birth (34–37 weeks)																
No 94.0% (47) ***	9.28 (7.57–11.01)	0.30	9.91 (9.01–10.84)	0.11	0.28 (0.18–0.44)	0.40	0.32 (0.22–0.49)	0.91	3.76 (3.30–4.38)	0.05	3.70 (3.11–4.44)	0.16	0.90 (0.74–1.11)	0.71	0.94 (0.81–1.10)	0.83
Yes 6.0% (3) ***	14.48 (11.24–18.65)		14.17 (11.40–17.60)		0.47 (0.15–1.48)		0.38 (0.12–1.20)		5.72 (5.64–5.81)		7.23 (4.00–13.07)		0.88 (0.38–2.00)		1.24 (0.55–2.83)	
Birth weight adequacy for gestational age																
AGA 75.6% (34) ***	9.37 (6.85–11.70)	0.88	10.12 (9.03–11.23)	0.73	0.31 (0.18–0.53)	0.27	0.36 (0.21–0.60)	0.31	3.67 (3.03–4.50)	0.92	3.71 (2.93–4.79)	0.52	0.93 (0.76–1.16)	0.11	1.00 (0.82–1.23)	0.27
SGA 11.1% (5) ***	10.01 (6.78–15.09)		9.60 (6.45–14.26)		0.13 (0.03–0.64)		0.14 (0.05–0.35)		3.62 (2.52–5.14)		3.30 (2.61–4.02)		0.52 (0.38–0.77)		0.68 (0.52–0.98)	
LGA 13.3% (6) ***	10.40 (6.53–15.80)		10.52 (7.53–13.68)		0.19 (0.05–0.62)		0.29 (0.066–0.88)		4.15 (3.31–5.20)		4.63 (3.17–8.24)		1.18 (0.55–3.99)		0.93 (0.59–1.46)	

Maternal blood *n* = 49 and umbilical cord blood *n* = 46; * arithmetic mean (±SD); ** median minimum–maximum); *** frequency (count); GM = geometric mean; CI = confidence Interval; R = Spearman’s correlation; AGA—appropriate for gestational age; SGA—small for gestational age; LGA—large for gestational age; ^a^ Spearman’s *p*-value; ^b^ Mann–Whitney-U-test *p*-value (gender, Apgar at 5 min, preterm birth); ^c^ Kruskal–Wallis test *p*-value (birth weight adequacy for gestational age); ^d^ for Apgar at 5 min < 8 there is no value for the concentration of metals in maternal blood and umbilical cord.

**Table 4 ijerph-19-04295-t004:** Maternal and birth characteristics between “not fail” and “fail” groups, Rio de Janeiro, Brazil, 2018.

Mother Characteristics	“Not Fail”	“Fail”	*p*-Value
Mother age (years) ^a^	29.53 ± 6.03	25.88 ± 7.87	0.08 ^d^
Per capita income (US$) ^a^	269.77 ± 174.78	241.68 ± 157.96	0.62 ^d^
Education (years) ^b^	14 (12–14)	14 (12–16)	0.75 ^e^
Ethnicity ^c^			
Non-white	22	13	0.75 ^f^
White	9	4	
Tobacco exposure ^c^			
No	17	13	0.14 ^g^
Yes	14	4	
Alcohol consumption ^c^			
No	15	9	0.69 ^g^
Yes	17	8	
Birth characteristics			
Birth weight (kg) ^a^	3.4 ± 0.6	3.2 ± 0.4	0.22 ^d^
Gestational age (weeks) ^b^	39.0 (38–40)	39 (38–39)	1.12 ^e^
Gender ^c^			
Male	22	9	0.28 ^g^
Female	10	8	
Apgar at 5 min ^c^			
≥8	31	17	1.00 ^f^
<8 ^c^	1	0	
Preterm birth (<37 weeks) ^c^			
No	31	15	0.27 ^f^
Yes	1	2	
Birth weight adequacy for gestational age ^c^			
AGA	19	14	0.52 ^f^
SGA	3	2	
LGA	5	1	

Maternal blood *n* = 49 and umbilical cord blood *n* = 46; AGA—appropriate for gestational age; SGA—small for gestational age; LGA—large for gestational age; ^a^ arithmetic mean ± sd; ^b^ median (Q1–Q3); ^c^ = n; ^d^ T test; ^e^ Mann–Whitney U-test; ^f^ Fisher test; ^g^ Chi-square test.

**Table 5 ijerph-19-04295-t005:** Geometric means of metal concentrations in maternal blood and cord blood of the “not fail” and “fail” groups in the DDST-II evaluation, Rio de Janeiro, Brazil, 2018.

		Arsenic µg/L				Cadmium µg/L				Lead µg/dL				Mercury µg/L			
		Maternal Blood		Cord Blood		Maternal Blood		Cord Blood		Maternal Blood		Cord Blood		Maternal Blood		Cord Blood	
GM *p*-value ^a^ (CI 95%)																	
Not fail 64.6% (31) *		8.47 (6.26–10.55)	0.03	9.51 (8.58–10.59)	0.14	0.28 (0.16–0.50)	0.99	0.33 (0.22–0.52)	0.64	3.73 (3.18–4.33)	0.84	3.57 (2.95–4.39)	0.73	0.93 (0.72–1.25)	0.69	1.02 (0.84–1.26)	0.33
Fail 35.4% (17) *		11.85 (9.39–15.24)		11.48 (9.53–13.49)		0.32 (0.13–0.80)		0.34 (0.14–0.84)		4.05 (3.10–5.55)		4.44 (3.18–6.56)		0.87 (0.66–1.23)		0.85 (0.66–1.15)	

Maternal blood *n* = 47 and umbilical cord blood *n* = 44; * frequency (count); GM = geometric mean; CI = confidence interval; DDST-II—Denver Development Screening Test-II; ^a^ Mann–Whitney U-Test.

**Table 6 ijerph-19-04295-t006:** Geometric means of metal concentrations in maternal blood and cord blood of the “not fail” and “fail” groups in the DDST-II tool for each assessed domain, Rio de Janeiro, Brazil, 2018.

	Arsenic µg/L				Cadmium µg/L				Lead µg/dL				Mercury µg/L			
	Maternal Blood		Cord Blood		Maternal Blood		Cord Blood		Maternal Blood		Cord Blood		Maternal Blood		Cord Blood	
GM *p*-value ^a^ (CI 95%)																
Personal social																
Not fail 84.0% (40) *	9.48 (7.62–11.70)	0.74	10.22 (9.17–11.42)	0.53	0.33 (0.20–0.55)	0.45	0.34 (0.21–0.53)	0.77	4.05 (3.48–4.74)	0.14	3.98 (3.26–4.90)	0.19	0.95 (0.76–1.23)	0.53	1.02 (0.86–1.23)	0.19
Fail 16.0% (8) *	9.75 (6.78–13.47)		9.74 (7.33–12.85)		0.13 (0.04–0.45)		0.33 (0.15–0.61)		2.75 (2.13–3.66)		3.11 (2.23–4.49)		0.69 (0.55–0.90)		0.65 (0.49–0.90)	
Fine motor adaptive																
Not fail 96.0% (48) *	9.37 (7.51–11.29)	0.27	9.99 (9.07–11.03)	0.33	0.28 (0.18–0.46)	0.86	0.33 (0.23–0.48)	0.91	3.91 (3.40–4.55)	0.17	3.91 (3.30–4.82)	0.13	0.91 (0.75–1.16)	0.30	0.97 (0.83–1.13)	0.31
Fail 4.0% (2) *	11.60 (11.41–11.79)		11.91 (10.88–13.04)		0.42 (0.08–2.32)		0.20 (0.01–3.95)		2.44 (2.37–2.52)		2.16 (1.69–2.77)		0.64 (0.62–0.67)		0.67 (0.56–0.80)	
Language																
Not fail 94.0% (47) *	9.31 (7.58–11.08)	0.36	9.94 (9.11–10.91)	0.33	0.29 (0.17–0.46)	0.77	0.33 (0.21–0.50)	0.48	3.84 (3.30–4.47)	0.48	3.86 (3.28–4.67)	0.59	0.92 (0.75–1.15)	0.77	0.97 (0.83–1.14)	0.15
Fail 6.0% (3) *	13.51 (9.79–18.65)		13.17 (9.86–17.60)		0.23 (0.15–0.35)		0.20 (0.12–0.34)		3.53 (2.21–5.64)		2.87 (2.07–4.00)		0.52 (0.39–0.70)		0.62 (0.55–0.70)	
Gross motor																
Not fail 80.0% (40) *	8.69 (6.96–10.51)	0.07	9.56 (8.63–10.51)	0.07	0.26 (0.15–0.41)	0.77	0.29 (0.19–0.45)	0.41	3.59 (3.11–4.11)	0.31	3.48 (2.97–4.16)	0.23	0.88 (0.72–1.14)	0.80	0.93 (0.79–1.09)	0.85
Fail 20.0% (10) *	12.75 (10.01–17.84)		12.06 (9.89–14.96)		0.41 (0.15–1.18)		0.43 (0.16–1.15)		4.77 (3.27–7.28)		5.21 (3.27–8.81)		0.95 (0.60–1.55)		1.04 (0.75–1.49)	

Maternal blood *n* = 47 and umbilical cord blood *n* = 44; * frequency (count); GM = geometric mean; CI = confidence interval; DDST-II—Denver Development Screening Test-II; ^a^ Mann–Whitney U-test *p*-value.

## Data Availability

The datasets generated during and/or analyzed during the current study are available from the corresponding author on reasonable request.

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
