# Peer review of "Prenatal Exposure to Metals and Neurodevelopment in Infants at Six Months: Rio Birth Cohort Study of Environmental Exposure and Childhood Development (PIPA Project)"

_ijerph, 2022, doi:10.3390/ijerph19074295_

Round 1
Reviewer 1 Report
General comments
This is an interesting and well-performed small study and is, to me, worth to be published. I have a number of mostly editorial comments (see below) that hopefully could help to improve the manuscript further. I have got only one general question that you should address. To me, it is absolutely not clear if your findings, in particular with regard to arsenic, indicate association or correlation. You could either say that further research is needed because of the rather small group size and the limited post-observation period of six months (as you did already in the Discussion section) or you could compare the data on neurobehavioural assessment of six-month old infants with information from published literature, from Brazil but also from other countries. To be honest, I have no idea what the rate of babies failing one or more categories in the DDST-II or similar tests usually is and how large the variability at an age of six months. Such data would enormously help to put your results into perspective.
Specific comments
Introduction, line 57: "... de Assis Araujo et al. (2020)" is the reference, of course. What does "(2020)19" mean? Most likely, the "19" should be in brackets.
In the following sentence, on lines 57 - 59, one might wonder how critical or hazardous lead concentrations of close to 5 µg/dL are. Before, you say that concentrations of four metals including Pb were above the LOD but this would not necessarily mean that there was a concern or risk. Then, you highlight lead because you give a more or less precise figure only for this metal. Or was lead the only one with concentrations above LOQ? Please clarify!
Material and methods, 2.1 Study area
Here, you briefly describe the role of UFRJ maternity in the public health system in Rio de Janeiro. I must confess that I find it difficult to catch the meaning of "reference" in this context. Is my understanding correct that women with high-risk pregnancies are directed to UFRJ maternity for medical care and to give birth there? Do women attending one of the eight familiy health centers usually deliver in this hospital? Or does it mean that UFRJ sets the "standards" for medical care of pregnant women? Please clarify since most readers abroad will not be familiar with health system and medical care in Brazil.
Figure 1: Should be "in the sixth month". Also: birth weight <2.5 kg (typo in manuscript).
2.3 Determinations
In line 110, a comma is lacking following 0.003 µg/L.
2.4 Screening test
It might be debatable whether the category "no opportunity" should be allocated to "not fail". Instead, in such event, the child might be better excluded.
2.5 Statistical analysis
Line 147: I have never seen p< 0.100. For the likelihood or error, p < 0.1 would be more usual. By the way, is that common for covariate analysis is a study of this type not to use a p value of p< 0.05?
2.6 Covariates
Line 155: English is not my mother tongue but "informed" should be, perhaps, replaced by "revealed" here since, apparently, the mother gave this information on request.
3. Results
Line 168: If the mean age of the mothers (at the day on which the gave birth, I assume) is given in a precise figure, the word "old" should be deleted from the sentence.
Line 169: Is the mean per capita income per month? (I guess.)
In the Tables 1 - 3, I have spotted repeatedly the word "valor". Is that perhaps the correct term in Portuguese but should be replaced, in this text, by "value"?
Table 3: is "Apgar 5" a common term in neonatal care or should it be briefly introduced? (I know only "Apgar", without a number, but more from my experience as a father than from my professional expertise.)
In the explanation below Table 3, there should be a space between "b" and "U". By the way, I think the more common name of the test is Mann-Whitney-U-test.
Line 200: Should be "Most (n=33, 66%)". My first understanding was that 33 to 66% of all toddlers did not fail and was a bit surprised about the rather high rate who failed. Later, it became clear what is meant here.
Below Table 4, it should be "chi-square" but not "chii" (Typo). Furthermore, there should be a space between lines 208 and 209.
Discussion, line 234: The numbers 26-29 should be in brackets.
Line 259: Should be Tolins et al. (2014). It is unusual to give three authors.
Line 270: Should be "toxic effects of this metal".
Line 274: I know what you mean and I completely agree and strongly support that you highlight the impact of socioeconomic status but you should check the sentence for appropriate English. Might be sufficient to say "including socioeconomically less favored ethnicities" or perhaps you could introduce socioeconomic status. As it is, it sounds a bit odd even though the meaning is clear.
With regard to lead you could discuss a bit more why no effects of higher lead concentratons were seen in your study. Of course, this would be a bit speculative, but an effect of lead on neurological development is what most scientists naturally would expect. Something about group size, selection bias or exposure in the Rio de Janeiro region? (In the introduction, you have emphasized lead concentrations elsewhere.)
Strength and limitations: Personally, when a I look at the results, I doubt that it would have been necessary here to take into account potential interactions, with all their uncertainties.
Author Response
Response to Reviewer 1 Comments
Dear reviewer
We thank you very much for the comments and suggestions. We are grateful for your time and constructive comments in our manuscript. We considered all your comments and suggestions.
Kind regards,
Mônica Seefelder de Assis Araujo
Master’s in Public Health
Federal University of Rio de Janeiro
General comments
This is an interesting and well-performed small study and is, to me, worth to be published. I have a number of mostly editorial comments (see below) that hopefully could help to improve the manuscript further. I have got only one general question that you should address. To me, it is absolutely not clear if your findings, in particular with regard to arsenic, indicate association or correlation. You could either say that further research is needed because of the rather small group size and the limited post-observation period of six months (as you did already in the Discussion section) or you could compare the data on neurobehavioural assessment of six-month old infants with information from published literature, from Brazil but also from other countries. To be honest, I have no idea what the rate of babies failing one or more categories in the DDST-II or similar tests usually is and how large the variability at an age of six months. Such data would enormously help to put your results into perspective.
Response General Comments: Our findings indicated an association between the exposure to higher arsenic concentration (above the geometric mean) and the occurrence of failure in the DDST – II test in our study population.
You could either say that further research is needed because of the rather small group size and the limited post-observation period of six months (as you did already in the Discussion section) or you could compare the data on neurobehavioural assessment of six-month old infants with information from published literature, from Brazil but also from other countries.To be honest, I have no idea what the rate of babies failing one or more categories in the DDST-II or similar tests usually is and how large the variability at an age of six months. Such data would enormously help to put your results into perspective.
Response: We agreed with your evaluation. We included this comparison in the manuscript Discussion section.
Specific Comments
Introduction, line 57: "... de Assis Araujo et al. (2020)" is the reference, of course. What does "(2020)19" mean? Most likely, the "19" should be in brackets.
Response Introduction, line 57: We added brackets to the “19”-
In the following sentence, on lines 57 - 59, one might wonder how critical or hazardous lead concentrations of close to 5 µg/dL are. Before, you say that concentrations of four metals including Pb were above the LOD but this would not necessarily mean that there was a concern or risk. Then, you highlight lead because you give a more or less precise figure only for this metal. Or was lead the only one with concentrations above LOQ? Please clarify!
Response lines 57 - 59:
All metals had blood concentrations detected above LD and LOQ, however, only lead has a health risk value for children exposure stablished. Since May 2021, the CDC health risk value to identify children with high lead levels is 3.5 μg/dL. [20]. For mercury, a cord blood limit of 5.8 μg/L is stablished by National Research Council (Ruggieri et al, 2017; NRC, 2000). We altered the sentences and added this information in the text.
Material and methods, 2.1 Study area
Here, you briefly describe the role of UFRJ maternity in the public health system in Rio de Janeiro. I must confess that I find it difficult to catch the meaning of "reference" in this context. Is my understanding correct that women with high-risk pregnancies are directed to UFRJ maternity for medical care and to give birth there? Do women attending one of the eight familiy health centers usually deliver in this hospital? Or does it mean that UFRJ sets the "standards" for medical care of pregnant women? Please clarify since most readers abroad will not be familiar with health system and medical care in Brazil.
Response Material and methods, 2.1 Study area:
The UFRJ Maternity is one of the public hospitals of Rio de Janeiro responsible for prenatal assistance and delivery to all pregnant women living in the city, including those with high-risk pregnancies. We rewrote this subsection.
Figure 1: Should be "in the sixth month". Also: birth weight <2.5 kg (typo in manuscript).
Response Figure 1: Corrected
2.3 Determinations
In line 110, a comma is lacking following 0.003 µg/L.
Response line 110: Corrected
2.4 Screening test
It might be debatable whether the category "no opportunity" should be allocated to "not fail". Instead, in such event, the child might be better excluded.
Response 2.4 Screening test: We agreed with your suggestion. We did new analysis and corrected the text and the tables 4, 5, and 6.
2.5 Statistical analysis
Line 147: I have never seen p< 0.100. For the likelihood or error, p < 0.1 would be more usual. By the way, is that common for covariate analysis is a study of this type not to use a p value of p< 0.05?
Response Line 147: We altered p value “p<0.100” for” p< 0.1”. To select which covariates should be included in a regression model, p < 0.05 is most often used, however for rigorous evaluation p value < 0.1 can be chosen.
2.6 Covariates
Line 155: English is not my mother tongue but "informed" should be, perhaps, replaced by "revealed" here since, apparently, the mother gave this information on request.
Response Line 155: We replaced “informed” by “revealed”.
- Results
Line 168: If the mean age of the mothers (at the day on which the gave birth, I assume) is given in a precise figure, the word "old" should be deleted from the sentence.
Response Line 168: We deleted the word "old" from the sentence.
Line 169: Is the mean per capita income per month? (I guess.)
Response Line 169: Yes, the mean per capita income is per month.
In the Tables 1 - 3, I have spotted repeatedly the word "valor". Is that perhaps the correct term in Portuguese but should be replaced, in this text, by "value"?
Response Tables 1 - 3: We checked the entire manuscript and replaced all “valor” terms by value.
Table 3: is "Apgar 5" a common term in neonatal care or should it be briefly introduced? (I know only "Apgar", without a number, but more from my experience as a father than from my professional expertise.)
Response Table 3- Apgar 5: Apgar 5 is common in neonatology, in Brazil, to identify the evaluation made at 5 minutes after delivery. For better understanding we changed it to Apgar at 5 minutes.
In the explanation below Table 3, there should be a space between "b" and "U". By the way, I think the more common name of the test is Mann-Whitney-U-test.
Response explanation below Table 3: We accepted your suggestion.
Line 200: Should be "Most (n=33, 66%)". My first understanding was that 33 to 66% of all toddlers did not fail and was a bit surprised about the rather high rate who failed. Later, it became clear what is meant here.
Response line 220: We corrected the sentence
Below Table 4, it should be "chi-square" but not "chii" (Typo). Furthermore, there should be a space between lines 208 and 209.
Response table 4: Corrected.
Discussion, line 234: The numbers 26-29 should be in brackets.
Response Discussion, line 234: Corrected
Line 259: Should be Tolins et al. (2014). It is unusual to give three authors.
Response Line 259: Corrected
Line 270: Should be "toxic effects of this metal".
Response Line 270: Corrected.
Line 274: I know what you mean and I completely agree and strongly support that you highlight the impact of socioeconomic status but you should check the sentence for appropriate English. Might be sufficient to say "including socioeconomically less favored ethnicities" or perhaps you could introduce socioeconomic status. As it is, it sounds a bit odd even though the meaning is clear.
Response Line 274: We agreed with your suggestion and corrected.
With regard to lead you could discuss a bit more why no effects of higher lead concentrations were seen in your study. Of course, this would be a bit speculative, but an effect of lead on neurological development is what most scientists naturally would expect. Something about group size, selection bias or exposure in the Rio de Janeiro region? (In the introduction, you have emphasized lead concentrations elsewhere.)
Response: We agreed with your evaluation. In fact, we would wait for a more expressive toxic effect of lead exposure on the neurodevelopment of our study population than related to the other metals exposure. One issue to be observed is that the As geometric mean (11.48 µg/L) is much higher than the Pb geometric mean (4.44 µg/dL) in the Fail group. It could be a reason but the small size of the group does not permit any conclusion. We included this observation in the manuscript Discussion section.
Strength and limitations: Personally, when a I look at the results, I doubt that it would have been necessary here to take into account potential interactions, with all their uncertainties.
Response Strength and limitations: We agreed with the reviewer. The metals interactions are the less critical factor in this pilot study. We removed this observation from the text.
Reviewer 2 Report
Revision – “Prenatal exposure to metals and neurodevelopment in infants at six months: Rio Birth Cohort Study of Environmental Exposure and Childhood Development (PIPA Project)”
Summary:
The authors of the present study analyzed the correlation of heavy metal exposure in pregnant women with the characteristics and neurodevelopment of their newborns in a cohort study based in Rio de Janeiro. The uptake of heavy metals e.g. by foods as well as their toxic effects are well known problems and were already studied for decades. However, the evaluation of these substances in more complex situations like pregnancy moves more into focus over the last years. Due to the known high toxicity of lead, arsenic, cadmium and mercury, it is important and necessary to evaluate the origin of these substances and their risks for newborns. The present study gives first insights in these correlations by identifying groups with higher risks of metal exposure as well as possibly induced effects, even if the study based on a regional small number of participants.
The introduction of the manuscript gives an overview about the issue and the need for further analyses. Nevertheless, for consideration of the results and suitable effects, the known toxic ranges of each metal should be given in the introduction or discussion section. The methods are sufficiently documented to allow replication or refer to corresponding articles. The used screening test was described in more detail, which makes it comprehensible for all reader. While the results are shown in several tables, structured by different characteristics (sociodemographic, blood type), each metal was considered individually in the discussion section. By doing so, the main results were well summarized and highlighted.
In conclusion, the present study should be published in IJERPH after revision of some minor points.
Minor points:
(1) Line 4: Please remove the point at the end of title
(2) Line 21: ‘One-third of the infants (17 – 33%) …’ Why is here a range or is 17 the total count? Please revise.
(3) Line 23: Remove the space in μg /L to μg/L. It should be consistent in the whole manuscript.
(4) Line 28: Please include keyword different from the title e.g. Denver Development Screening Test II
(5) Line 37: ‘… toxic human health effects.’ Please give some examples.
(6) Line 57: ‘… Araujo et al. (2020)19…’ Please remove 19 or was it published in 2019?
(7) Line 59: ‘… Pb concentrations near 5μg/dL…’ Please add if such a concentration is relatively high or low and include the toxic concentration of the metals. Otherwise the results cannot estimated.
(8) Line 60: Remove ‘and’.
(9) Line 76: The study was carried out from September to August. In the abstract and in line 83 you mentioned October. Please clarify.
(10) Line 82 (Study population): Mothers using drugs during pregnancy were excluded from the study. Why do you not exclude the use of tobacco and alcohol? Even if you not see a correlation between alcohol and the neurodevelopmental tests, these substances might influence the results.
(11) Line 93 (Figure 1): The figure is great, but need to be revised. Due to formatting the numbers and texts are partly not visible.
(12) Line 141-142: Please define the continuous and categorical variables.
(13) Line 162 (Table 1): Why are the values of P95 lower than P90? That should not be possible.
Please revise. Moreover, the term p value instead of p valor should be used in the whole
manuscript.
(14) Line 165: SD is the abbreviation for standard deviation. Did you use the standard deviation
(SD) or the standard error mean (SEM)? Please clarify.
(15) Line 166: The figure legend says “Lead 0.05 μg/L” while in the table you use μg/dL. It should
be consistent.
(16) Table 2, 3, 5 and 6: The tables are well structured, but it is confusing to see values over two
rows. May you can show the tables in landscape format or you may split the table in Arsenic +
Cadmium and Lead + Mercury.
(17) Line 219: As similar to the other metals mercury should be written bold.
(18) Line 234: use brackets for the citation numbers.
(19) Line 248 (Arsenic): You were able to detect a significant difference of Arsenic in DDST-II
evaluation (Table 5). These results are in line with the results of the Gross motor results (Table
6). Even if you do not see a significant difference (0.07), there is a tendency and this tendency
should be mentioned in the discussion section. This will highlight your results.
(20) Line 293: ‘… high child exposure …’ Could you please include the measured concentration and
compare them with your results.
(21) Line 333 (References): Please revise the formatting. The publishing year partly stands at the
end of the reference, but partly it follows after the author names (e.g. reference number 33).
Author Response
Response to Reviewer 2 Comments
Dear reviewer
We thank you very much for the comments and suggestions. We are grateful to for your time and constructive comments in our manuscript. We considered all your comments and suggestions.
Kind regards,
Mônica Seefelder de Assis Araujo
Master’s in public health
Federal University of Rio de Janeiro
Comments and Suggestions for Authors
Minor points:
(1) Line 4: Please remove the point at the end of title
Response Point (1): Corrected.
(2) Line 21: ‘One-third of the infants (17 – 33%) …’ Why is here a range or is 17 the total count? Please revise.
Response point (2): “17” is the total count. We altered (17 – 33%) to (n= 17 – 35.4%). The percent value was changed because we attended a suggestion fromReviewer 1: “It might be debatable whether the category "no opportunity" should be allocated to "not fail". Instead, in such event, the child might be better excluded.”
(3) Line 23: Remove the space in μg /L to μg/L. It should be consistent in the whole manuscript.
Response point (3): Corrected.
(4) Line 28: Please include keyword different from the title e.g., Denver Development Screening Test II
Response point (4): We included “neurodevelopmental screening”.
(5) Line 37: ‘… toxic human health effects.’ Please give some examples.
Response point (5): We rewrote this sentence “Chronic low-dose exposure to these metals may cause toxic human health effects affecting the nervous, hematological and immune systems”
(6) Line 57: ‘… Araujo et al. (2020)19…’ Please remove 19 or was it published in 2019?
Response point (6): “19” is the reference number, we added brackets to the “19”.
(7) Line 59: ‘… Pb concentrations near 5μg/dL…’ Please add if such a concentration is relatively high or low and include the toxic concentration of the metals. Otherwise the results cannot estimated.
Response point (7): Since May 2021, the CDC health risk value to identify children with high lead levels is 3.5 μg/dL. [20]. For mercury, a cord blood limit of 5.8 μg/L is stablished by National Research Council (Ruggieri et al, 2017; NRC, 2000). We altered the sentences and added this information in the text.
(8) Line 60: Remove ‘and’.
Response point (8): We removed “Corrected”.
(9) Line 76: The study was carried out from September to August. In the abstract and in line 83 you mentioned October. Please clarify.
Response point (9): The entire study, including participant capture, data collection and biological samples, deliveries, and follow-up, was carried out from September 2017 to August 2018. The babies were born between October 2017 and February 2018
(10) Line 82 (Study population): Mothers using drugs during pregnancy were excluded from the study. Why do you not exclude the use of tobacco and alcohol? Even if you not see a correlation between alcohol and the neurodevelopmental tests, these substances might influence the results.
Response point (10): We chose to keep babies whose mothers consumed alcohol or were exposed to tobacco, as both are confounding variables widely described in the literature. To control these confounding variables, we evaluated the distribution of tobacco and alcohol mother´s consumption between the Fail and not Fail groups, and we did not find a significant difference.
(11) Line 93 (Figure 1): The figure is great, but need to be revised. Due to formatting the numbers and texts are partly not visible.
Response point (11): We corrected the figure 1.
(12) Line 141-142: Please define the continuous and categorical variables.
Response point (12): Continuous variables were obtained by measuring or counting: mother age, gestational age, birth weight, per capita income, education counted in years.
Categorical variable are qualitative data used to set groups according to some individual characteristics (ethnicity, gender), lifestyle (alcohol consumption, tobacco exposure) or any kind of classification (Apgar, birth weight adequacy for gestational age, prematurity)
We added this information in Covariates subsction
(13) Line 162 (Table 1): Why are the values of P95 lower than P90? That should not be possible. Please revise. Moreover, the term p value instead of p valor should be used in the whole manuscript.
Response point (13): We revised and corrected the table 1. We checked the entire manuscript and replaced all “valor” terms by value.
(14) Line 165: SD is the abbreviation for standard deviation. Did you use the standard deviation (SD) or the standard error mean (SEM)? Please clarify.
Response point (14): We used standard deviation. We corrected this item.
(15) Line 166: The figure legend says “Lead 0.05 μg/L” while in the table you use μg/dL. It should be consistent.
Response point (15): We corrected this item.
(16) Table 2, 3, 5 and 6: The tables are well structured, but it is confusing to see values over two rows. May you can show the tables in landscape format or you may split the table in Arsenic +Cadmium and Lead + Mercury.
Response point (16): We changed the tables to landscape format.
(17) Line 219: As similar to the other metals mercury should be written bold.
Response point (17): At the editor's suggestion we have removed all bolds from the tables.
(18) Line 234: use brackets for the citation numbers.
Response to point (18): Corrected
(19) Line 248 (Arsenic): You were able to detect a significant difference of Arsenic in DDST-II evaluation (Table 5). These results are in line with the results of the Gross motor results (Table 6). Even if you do not see a significant difference (0.07), there is a tendency and this tendency should be mentioned in the discussion section. This will highlight your results.
Response point (19): We included this observation in the discussion section.
(20) Line 293: ‘… high child exposure …’ Could you please include the measured concentration and compare them with your results.
Response point (20): The studies reported by Asmus et al. measured mercury concentrations in hair; that is why we are not comparing them with our results. We could not find other Brazilian studies that had analysed metals concentrations in cord blood.
(21) Line 333 (References): Please revise the formatting. The publishing year partly stands at the end of the reference, but partly it follows after the author names (e.g. reference number 33).
Response point (21); We reviewed the references formatting.

Round 2
Reviewer 1 Report
Thank you for taking my considerations into account. I agree with all your conclusions and with the revisions that you have made. As I expected, the minor revisions in the manuscript have improved it. The only point that I would like you to ask you to check is Figure 1. You have amended there a new phrase on children who could not been evaluated due to opportunity. (I guess that this is what you wanted to write.) There are typos (evalueted, opotunity) and, perhaps, "no opportunity" should be explained.
Author Response
Response to Reviewer 1 Comments
Dear reviewer
Thank you again for the comments and suggestions.
Kind regards,
Mônica Seefelder de Assis Araujo
Master’s in public health
Federal University of Rio de Janeiro
Thank you for taking my considerations into account. I agree with all your conclusions and with the revisions that you have made. As I expected, the minor revisions in the manuscript have improved it. The only point that I would like you to ask you to check is Figure 1. You have amended there a new phrase on children who could not been evaluated due to opportunity. (I guess that this is what you wanted to write.) There are typos (evalueted, opotunity) and, perhaps, "no opportunity" should be explained.
Response:
Typo corrected.
We explained “no opportunity” at the subsection 2.2 Study Population, second paragraph, lines 92-95.
